# Material Removal Capability and Profile Quality Assessment on Silicon Carbide Micropillar Fabrication with a Femtosecond Laser

**DOI:** 10.3390/ma16010244

**Published:** 2022-12-27

**Authors:** Xifang Zhang, Zhibao Hou, Jiacheng Song, Zhiyi Jin, Zhenqiang Yao

**Affiliations:** 1School of Mechanical Engineering, Shanghai Jiao Tong University, Shanghai 200240, China; 2State Key Laboratory of Mechanical System and Vibration, Shanghai Jiao Tong University, Shanghai 200240, China

**Keywords:** silicon carbide (SiC), femtosecond laser, micropillars, profile quality, material removal capability (MRC)

## Abstract

Silicon carbide (SiC) has a variety of applications because of its favorable chemical stability and outstanding physical characteristics, such as high hardness and high rigidity. In this study, a femtosecond laser with a spiral scanning radial offset of 5 μm and a spot radius of 6 μm is utilized to process micropillars on a SiC plate. The influence of pulsed laser beam energies and laser translation velocities on the micropillar profiles, dimensions, surface roughness *R*a, and material removal capability (MRC) of micropillars was investigated. The processing results indicate that the micropillar has the best perpendicularity, with a micropillar bottom angle of 75.59° under a pulsed beam energy of 50 μJ in the range of 10–70 μJ, with a pulsed repetition rate of 600 kHz and a translation velocity of 0.1 m/s. As the laser translation velocity increases between 0.2 m/s and 1.0 m/s under a fixed pulsed beam energy of 50 μJ and a constant pulsed repetition rate of 600 kHz, the micropillar height decreases from 119.88 μm to 81.79 μm, with the MRC value increasing from 1.998 μm^3^/μJ to 6.816 μm^3^/μJ, while the micropillar bottom angle increases from 68.87° to 75.59°, and the *R*a value diminishes from 0.836 μm to 0.341 μm. It is suggested that a combination of a higher pulsed laser beam energy with a faster laser translation speed is recommended to achieve micropillars with the same height, as well as an improved processing efficiency and surface finish.

## 1. Introduction

Silicon carbide (SiC) has been recognized as a promising material with excellent characteristics for use in mechanical seals, sliding bearings, precision molds, and cutting tools owing to its high stiffness, hardness, and chemical stability [1,2]. SiC has low specific gravity, wide bandgaps, high thermal conductivity, and low thermal expansion, in addition to merits as a third-generation semiconductor [3,4]. SiC can be widely applied in optical devices, nanotechnology, and nuclear material science [5,6]. Due to its extraordinarily biocompatibility and chemical inertness, SiC has been used in power systems, including those related to driving, lighting, and uninterrupted power supplies [7,8,9]. The innovation of SiC surface treatments was motivated because of the increasing requirements for high-performance components and low-cost automation systems. The surfaced texturing technique is an effective way to generate specific pillars, grooves, or dimples on the workpiece surface to significantly improve the performance of devices used under high temperatures and in extreme environments. SiC fabricated with microstructures can exhibit improved surface characteristics, including wettability in lubrication, reflectivity in optical applications, and endurability in sliding [10,11,12].

A number of technologies are available for surface texturing, such as micromilling, abrasive jet machining, molding, ultrasonic aided machining, electrochemical machining and electrical discharge machining [13,14]. Micromachining with lasers on metals, semiconductors, polymers, and other materials has become increasingly popular in many fields [15]. Three kinds of lasers—a CW laser, a pulsed beam with nanosecond duration, or an ultrafast pulsed beam with picosecond or even femtosecond duration—can be utilized to produce a surface geometry with high resolution by means of laser ablation as noncontact processing [16]. Compared to the long-pulsed laser width, the material removal mechanism could be conducted closer to the conditions of cold processing by means of a femtosecond laser, which can achieve micro/nanoprocessing accuracy in geometry with a negligible thermal affects zone, thanks to the ultrashort pulse width and ultrahigh instantaneous power [17]. Because of the high bonding strength inside SiC with covalent bonds, it is difficult to obtain various micro/nanostructures on the SiC surface using traditional processing approaches [18,19]. Femtosecond laser machining is an ideal processing technology for achieving surface textures with high processing accuracy on brittle materials with high hardness.

The femtosecond laser processing of textures on SiC has attracted worldwide attention and has been proven to have potential research value due to the utilization of SiC in various important areas of science and engineering. The microscopic mechanism in femtosecond laser processing of SiC was investigated with the two-temperature model, considering both the carrier concentration and the temperature change in SiC resulting from femtosecond laser irradiation [20]. The laser-induced periodic structure of SiC resulting from femtosecond laser ablation is mainly dominated by material lattice cleavage and influenced by the intrinsic texture, while the ablation threshold depends on both the laser pulse number and the processing medium, such as water, to achieve a smoother surface finish without oxidation on the SiC surface [21]. The microgrooves can be obtained with a femtosecond laser to achieve a high aspect ratio of approximately 82.67% on SiC by optimizing the process parameters of laser polarization direction, SiC crystal orientation, multiple-irradiation translation, and in depth feed rate during femtosecond laser machining [22]. A square sensor cavity diaphragm on 4 H-SiC with dimensions of 5 × 5 mm and a depth of 70 μm was generated by means of the etching method with a Nd/YVO4 laser [23]. Femtosecond laser and fluorine-based reactive ion etching can be combined as a new processing approach for micro/nano texturing on porous SiC, inducing significant surface morphology modification and tuning the optical properties of SiC as selective solar absorbers. The surface features could be adjusted by tailoring the scanning velocity and pulse frequency for laser radiation and radio frequency-power, as well as the mixture of gas and its pressure, to optimize reactive ion etching [24].

The present work investigates the process of micropillar formation on SiC using femtosecond laser scanning in a spiral mode. The impacts of pulsed beam energies and pulsed translation velocities on the micropillar profiles, dimensions, micropillar bottom angle, surface roughness, and material removal capability were examined.

## 2. Experimental Procedures

### 2.1. Fabrication of Micropillars on SiC with a Femtosecond Laser

As shown in Figure 1, a femtosecond laser processing system was set up to achieve micropillar arrays on an SiC substrate. The SiC workpiece was located on the workbench and monitored with a charge-coupled device for the laser spot of focus on the SiC surface. The laser processing path and parameters were scheduled and implemented through the computer control program. A femtosecond laser with a maximum pulsed beam energy of 400 μJ, a wavelength of 1028 nm, a pulse duration of 218 fs, and a base pulse repetition rate of 1 MHz was selected as the power supply (Carbide-40 W, Light Conversion, Vilnius, Lithuania). A galvanometer was used to control the scanning path to process the micropillars on the SiC plate.

The processing parameters for the femtosecond laser to obtain micropillars on a SiC substrate are listed in Table 1. A focused spot radius of 6 μm in the laser beam and a laser pulse repetition rate of 600 kHz were chosen to process the workpiece of SiC via 10 spiral scanning cycles. The laser processing parameters were scheduled from the combination of different laser pulsed beam energies in the range of 10–70 μJ and laser translation speeds varying between 0.2 m/s and 1.0 m/s.

To achieve a micropillar with a diameter of 300 μm, the femtosecond laser scanned in a helical approach with a radial thread interval of 5 μm for 10 cycles, resulting in a blind ring with a starting diameter *d*_1_ of 300 μm and a finished diameter *d*_2_ of 400 μm around the micropillars, as shown in Figure 2. The micropillar array of blind rings was processed with a pitch of 500 μm in the row and column directions. When the processing range is more than 60 mm over the interval of the galvanometer, the worktable to fix the SiC specimen will be used for positioning the micropillars.

Regarding the affecting depth of the laser beam, a Rayleigh length of approximately 110 μm can be predicted according to the laser wavelength, as well as Gaussian beam radius [25] from Table 1, corresponding to a depth of field of approximately 220 μm, indicating that vertical feeding is not needed for processing micropillars with heights less than 220 μm. The profiles of the micropillars were examined using a three-dimensional (3 D) profilometer (VK-X3000, Keyence, Osaka, Japan). The surface topographies of the micropillars on SiC were detected utilizing a scanning electron microscope (SEM, VEGA3 TESCAN, Brno, Czech Republic). The micropillar with a tapered form is directly related to the femtosecond laser radiation variables, where the upper appearances and bottom appearances were primarily affected by the pulse beam energies and transition velocities. The micropillar bottom angle, shown in Figure 3, can be indicated as follows [26]:(1)$$\alpha =\tan^{-1}[\frac{2H}{D_{1}-D_{2}}]$$
where *α* is the micropillar bottom angle, *H* represents the height of the micropillar, *D*_2_ denotes the upper diameter, and *D*_1_ denotes the bottom diameter determined with pulsed laser beam energy and transition velocity. A larger micropillar bottom angle indicates a better perpendicularity of the micropillars.

### 2.2. Material Removal Capability of the Femtosecond Laser

The material removal capability (MRC) demonstrates the capacity of the material removal volume subjected to the pulsed laser beam energy, which also implies the femtosecond laser processing efficiency. The MRC is expressed in terms of *ƞ* as follows:(2)$$\eta =\frac{V}{Q}$$
where *V* represents the material removal volume, and *Q* is the total pulse energy applied to the material during the processing time. The material removal volume, *V*, can be derived from:(3)$$V=\frac{\pi ({d_{2}}^{2}-{d_{1}}^{2})}{4}*H$$
where *d*_1_ is the starting diameter, *d*_2_ is the finished diameter of the blind ring after spiral scanning, and *H* is the height of the micropillar. The total pulse energy applied to the processed material, *Q*, can be denoted as:(4)Q=E∗f∗t
where *E* is the single pulsed beam energy, *f* is the pulse frequency, and *t* denotes the whole processing time.

Since the laser scanning path is an isometric helix with an offset of 5 μm, the machining time, *t*, can be approximately calculated as:(5)$$t=\frac{\pi (d_{1}+d_{2})}{2\nu }*n$$
where ν represents the laser translation speed, and *n* is the scanning cycles.

According to Equation (2) to Equation (5), the material removal capability can be deduced as follows:(6)$$\eta =\frac{(d_{2}-d_{1})*\nu *H}{2E*n*f}$$

Therefore, the MRC is closely related to the laser translation velocity, pulsed beam energy, and pulsed repetition rate.

## 3. Results and Discussion

### 3.1. Effect of Pulsed Laser Beam Energy on Micropillar Processing

To explore the influence of femtosecond pulsed beam energies on the material removal capability, micropillar geometric profiles, and surface roughness, micropillar arrays were produced with pulsed beam energies in the range of 10–70 μJ, with a constant laser translation velocity of 1.0 m/s and the same pulse repetition rate of 600 kHz. Under each pulsed beam energy, 9 micropillars were fabricated, and the profile errors were measured. The mean height of the micropillar array generated with different laser pulse energies is illustrated in Figure 4a, where the mean height increases from 18.33 μm to 101.93 μm, and the height variance of 9 micropillars varies from 0.61 to 1.08 μm, with the relative variation ranging from 0.03 to 0.01 as the pulsed laser beam energy varies from 10 μJ to 70 μJ, with 10 cycles of helical tracing of the laser beam around every micropillar. Figure 4b indicates the linear fitting of micropillar height against laser pulse energy, with a variation rate of 1.463 and an intercept of 5.085, showing a constant increment of micropillar height with the laser pulse energy. This is attributed to more laser ablation energy acting on the SiC substrate with increasing pulsed laser beam energy. When the pulsed laser beam energy exceeds 50 μJ, the increase in micropillar height resulting from the increase in pulsed laser beam energy starts to fade.

Figure 5 shows the variations in both the mean upper and bottom diameters for micropillar arrays under different laser pulse energies. When the pulsed laser beam energy varies from 10 μJ to 70 μJ, the upper diameter shrinks from 281.67 μm to 264.55 μm, and the bottom diameter expands from 311.36 μm to 318.11 μm. The differences in bottom diameters and upper diameters increase with increasing pulsed beam energies. When the pulsed laser beam energy exceeds 50 μJ, the diameter difference between the upper and bottom circles reaches a higher value of more than 49 μm.

The influence of pulsed beam energies on the micropillar bottom angle can be seen in Figure 6, where the bottom angle increases from 51.0° to 75.59° as the pulsed laser beam energy increases from 10 μJ to 50 μJ and saturates from 50 μJ to 70 μJ as the bottom angle decreases slightly from 75.59° to 75.28°. The micropillar bottom angle obtains the largest value of 75.59° under a pulsed beam energy of 50 μJ, which results from the fact that a laser pulse energy of 50 μJ brings about a higher increment (1.71 μm/μJ) in the micropillar height, as shown in Figure 4, while the increasing rate remains constant (1.01 μm/μJ) from 50 μJ to 70 μJ. On the other hand, a smaller diameter difference between the upper and bottom cross sections is demonstrated in Figure 5. This is attributed to the fact that the material removal capability decreases with increasing pulsed beam energy, as shown in Figure 7, where the declining rate remains the same (0.04 μm^3^/μJ^2^) from 50 μJ to 70 μJ, resulting in a narrower valley bottom and therefore, a larger micropillar bottom diameter, as shown in Figure 8. Both the increase in the micropillar height at a constant rate and the increase in the micropillar bottom diameter, due to the declining material removal capability, account for the saturation of the micropillar bottom angle, as the pulsed beam energy varies from 50 μJ to 70 μJ.

Three-dimensional (3 D) microscope images and geometric profiles for various laser beam energies with a constant laser translation velocity of 1.0 m/s are displayed in Figure 8. Spiral scanning will bring about a shallow blind ring, with a larger fillet radius under a lower pulsed laser beam energy, but a deeper blind ring with a smaller fillet radius and a steeper sidewall under a higher pulsed laser beam energy.

The impacts of pulsed beam energies on the MRC and surface roughness *R*a value are exhibited in Figure 7, with a laser translation speed of 1.0 m/s and a pulse repetition rate of 600 kHz. The MRC value declines gradually from 7.637 μm^3^/μJ to 6.067 μm^3^/μJ, and the *R*a value increases continuously from 0.219 μm to 0.502 μm as the pulsed laser beam energy varies from 10 μJ to 70 μJ. The pulsed beam energy of 50 μJ separates both the MRC variation rate and *R*a variation rate. When the pulsed beam energy exceeds 50 μJ, the MRC value declines rapidly, and the *R*a value increases quickly.

This can be due to both the Gaussian energy distribution degradation from the focal aperture plane in the direction of the micropillar height, leading to the processing efficiency declining with the increase in the micropillar height, and more heat deposition at the bottom of the processing area with the layer-by-layer spiral scanning path when the pulsed laser beam energy increases from 10 μJ (a shallow micropillar) to 70 μJ (a deep micropillar), while the laser translation speed remains steady at 1.0 m/s, with a fixed translation rate of 600 kHz, leading to the surface morphology being coarser. In addition, the rapid deposition of melting sputtered materials on the micropillar bottom [27] accounts for the MRC value decreasing and the *R*a value increasing with increasing micropillar height.

### 3.2. Effect of Laser Translation Velocity on Micropillar Processing

The laser translation velocity is another parameter correlated with the geometric profiles, micropillar bottom angle, MRC, and *R*a value of micropillars processed with a femtosecond laser. The micropillars obtained with the laser translation velocity varied between 0.2 m/s and 1.0 m/s, with a fixed pulse laser beam energy of 50 μJ and the same pulse frequency of 600 kHz. Figure 9a indicates that the mean height of the micropillars drops from 119.88 μm to 81.79 μm as the laser translation velocity changes in the range of 0.2–1.0 m/s. Figure 9b illustrates the linear fitting of the micropillar height against the laser translation velocity, with a variation rate of −0.048 and an intercept of 129.852, showing a constant decrease in the micropillar height with the laser translation velocity. Less pulsed beam energy applied to the SiC surface due to shorter processing time over the 10 scanning rounds at a higher laser translation speed accounts for the decrease in micropillar heights.

The variations in both the mean upper and bottom diameters for micropillar arrays at different laser translation speeds are exhibited in Figure 10, where the upper diameter increases, but the bottom diameter decreases with increasing laser translation speed. When the laser translation velocity is in the range of 0.2–1.0 m/s, the upper diameter increases in the range of 261.73–271.56 μm, and the bottom diameter decreases from 354.41 μm to 313.59 μm. This results from the material removal mechanism at which higher translation speeds, laser energy would be applied to a larger processing area with a smaller thickness on the SiC workpiece surface. The high translation velocity brings about a larger material removal rate in the blind ring; hence, the small fillets and bottom diameters of micropillars remain.

Figure 11 indicates the variation in the bottom angle with increasing laser translation velocity, where a high translation velocity accompanies a large micropillar bottom angle with better perpendicularity. Due to the spiral scanning approaches, the micropillars generated were cone shaped. Moreover, the emergence of the truncated corners would increase the diameters of the bottom. At a higher laser translation speed, laser energy would be applied to a larger processing area, with a smaller thickness on the SiC workpiece surface, which helps more of the SiC surface to absorb laser energy sufficiently, resulting in a higher MRC and a smaller heat affecting thickness, leading to a lower *R*a.

Figure 12 displays the impacts of laser translation velocity on the MRC and *R*a values, with a constant pulsed laser beam energy of 50 μJ and the same pulse frequency of 600 kHz. It indicates that the MRC value increases from 1.998 μm^3^/μJ to 6.816 μm^3^/μJ, whereas the *R*a value diminishes from 0.836 μm to 0.341 μm as the laser translation velocities vary in the range of 0.2–1.0 m/s. It can be known from Equation (6) that the MRC is positively correlated with the laser translation speed and micropillar height at a constant pulsed laser beam energy and pulsed repetition rate. The offset between two conterminous pulsed laser beams ranges from 0.33 to 1.67 μm, according to the pulse frequency of 600 kHz, and the laser translation velocities vary in the range of 0.2–1.0 m/s, which is within the laser spot radius of 6 μm, indicating an obvious overlap of conterminous laser spots. The higher laser translation velocity will bring about less overlap, resulting in more material removal volume from the current layer and less material removal volume from the next layer. Since the material removal capability corresponds to the distance from the focused laser height, a higher laser translation velocity means a higher material removal capability. With decreasing laser translation speed, more heat is accumulated at the bottom of the blind ring around the micropillar, leading to a rougher surface finish, as exhibited in Figure 13. This shows that there is a rough profile on the top appearance of the micropillar at a laser translation velocity of 0.2 m/s. The surface quality can be improved with a faster laser translation speed at the same pulsed laser beam energy and pulse frequency.

### 3.3. Optimum Parameter Combination to Enhance Processing Efficiency and Surface Finish

The above experiments verified that both the pulsed laser beam energy and laser translation speed have significant effects on the processing efficiency and surface finish of micropillars at the same pulsed repetition rate. The question still remains as to which combination of these parameters is better for enhancing processing efficiency and surface finish: a lower pulsed laser beam energy with slower laser translation speed or a higher pulsed laser beam energy with faster laser translation speed.

An additional two processing experiments were carried out to examine the effects of different combinations of pulsed beam energies and laser translation velocities on the micropillar bottom angle, MRC, and *R*a at the same micropillar height of 92 μm and 102 μm, respectively, as shown in Table 2 and Table 3. The quality parameters indicate that for micropillars of the same height, the use of a higher pulsed laser beam energy with a faster laser translation speed could achieve micropillars with a larger bottom angle, higher material removal capability, and lower surface roughness. The geometric profile and SEM image of the micropillars fabricated at the same heights of 92 μm and 102 μm, with different combinations of laser machining parameters, are displayed in Figure 14 and Figure 15, respectively. It can clearly be shown that there is a better surface quality for micropillars generated with a higher pulsed beam energy with a faster laser translation speed than for those generated with a lower pulsed laser beam energy with a slower laser translation speed. Therefore, a higher pulsed beam energy with a faster laser translation speed is a better combination of laser ablation parameters to enhance processing efficiency and surface finish.

## 4. Conclusions

The processing of micropillars on SiC by utilizing a femtosecond laser with a helical scanning strategy was explored. According to the investigations of the influence of pulsed beam energies and laser translation speeds on the micropillar bottom angle, material removal capability, and surface roughness, the conclusions can be demonstrated as follows:

(1) The micropillar height increases constantly with a variation rate of 1.463 as the pulsed laser beam energy increases from 10 μJ to 70 μJ at the same laser translation velocity of 1.0 m/s and a constant pulsed repetition rate of 600 kHz, while the material removal capability (MRC) declines slightly from 7.637 μm^3^/μJ to 6.067 μm^3^/μJ, and the *R*a value increases continuously from 0.219 μm to 0.502 μm. When the pulsed laser beam energy exceeds 50 μJ, the increase in micropillar height resulting from the increase in pulsed laser beam energy starts to fade. The micropillar could obtain the best perpendicularity with a bottom angle of 75.59° at a pulsed laser beam energy of 50 μJ.

(2) As the laser translation velocities vary in the range of 0.2–1.0 m/s at the same pulsed beam energy of 50 μJ and a constant pulsed repetition rate of 600 kHz, the micropillar height decreases with a variation rate of −0.048 and an intercept of 129.852, while the MRC value increases obviously from 1.998 μm^3^/μJ to 6.816 μm^3^/μJ, the bottom angle increases from 68.87° to 75.59° with a better perpendicularity, and the *R*a value diminishes from 0.836 μm to 0.341 μm, with a better surface finish. It is suggested that the higher laser translation speed with correspondingly increased MRC value will bring about both a better surface finish and a better micropillar perpendicularity, resulting from the material removal mechanism in which the laser energy would be applied to a larger processing area, with a smaller thickness, on the SiC workpiece surface at a higher laser translation speed.

(3) To achieve micropillars with the same height using the proposed spiral scanning approach, a combination of a higher pulsed laser beam energy with a faster laser translation speed is recommended to improve the processing efficiency and surface finish.

## Figures and Tables

**Figure 1 materials-16-00244-f001:**
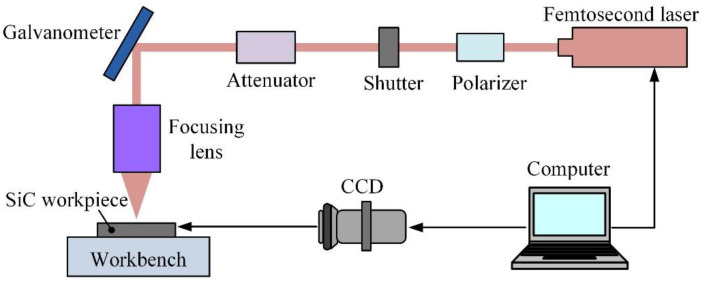
The setup of the femtosecond laser processing system for SiC surface treatment.

**Figure 2 materials-16-00244-f002:**
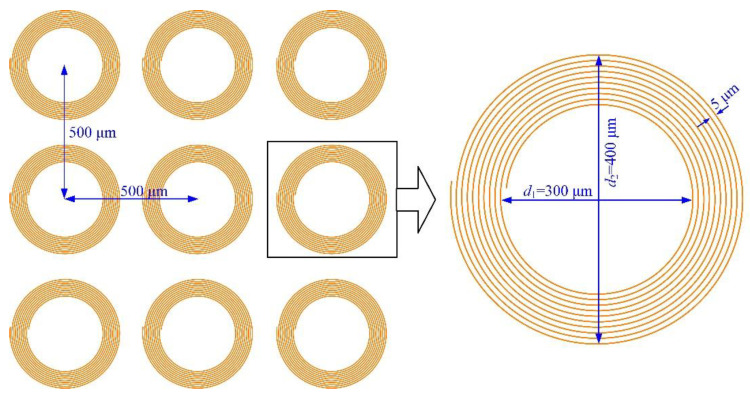
Spiral scanning approach to process SiC micropillars with a femtosecond laser.

**Figure 3 materials-16-00244-f003:**
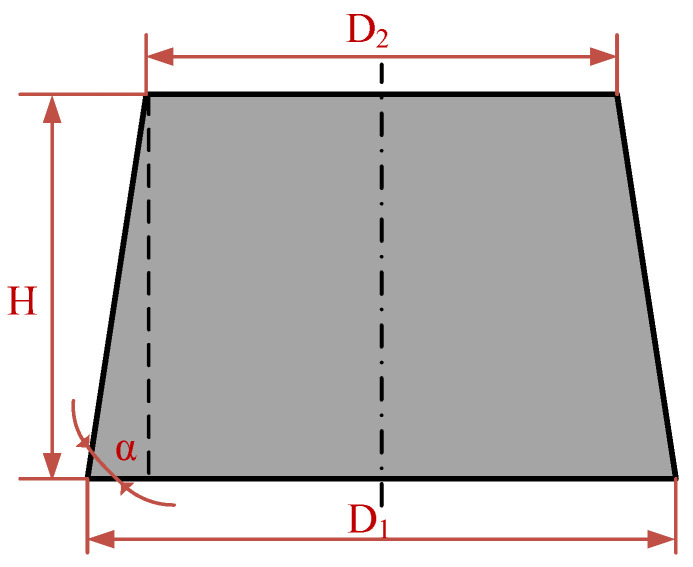
Diagrammatic drawing of the micropillar bottom angle.

**Figure 4 materials-16-00244-f004:**
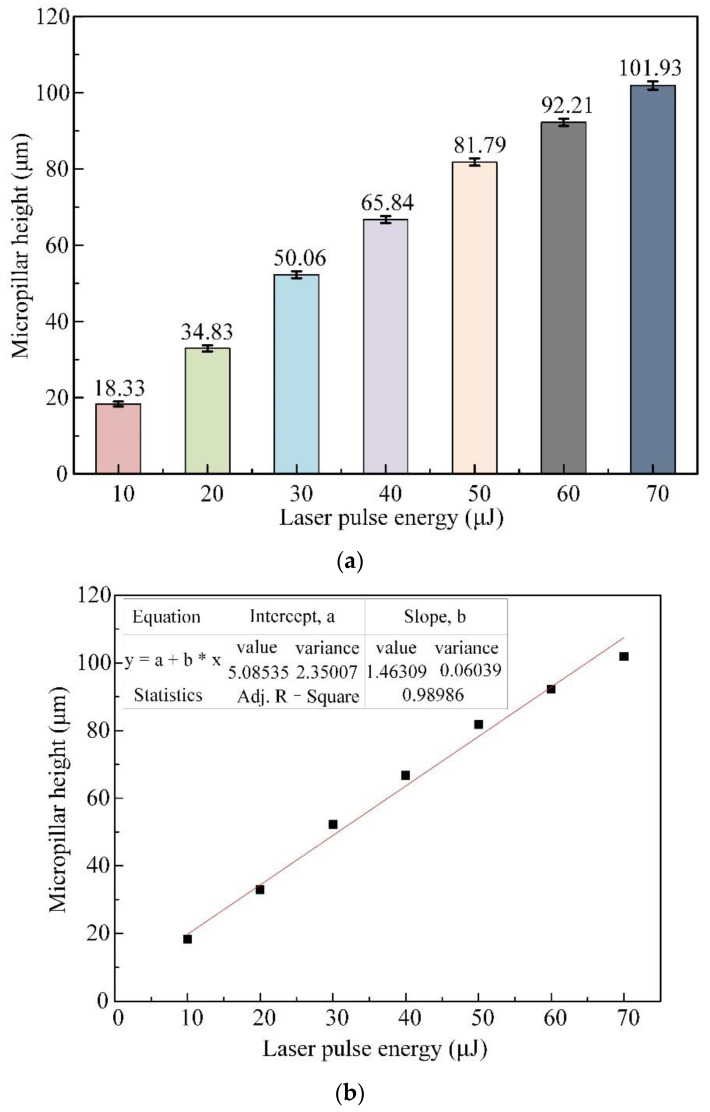
Effect of pulsed beam energies on the mean height of micropillars with a translation velocity of 1.0 m/s and a pulsed repetition rate of 600 kHz. (**a**) Measurements of micropillar height against laser pulse energy. (**b**) Linear fitting of micropillar height against laser pulse energy.

**Figure 5 materials-16-00244-f005:**
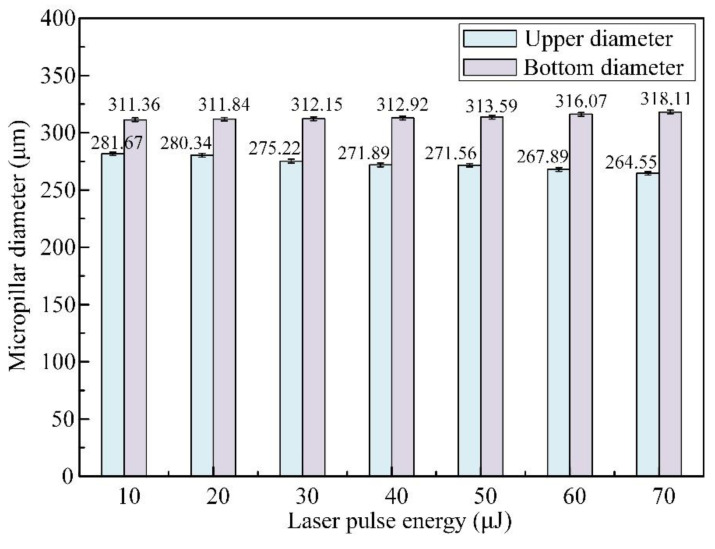
Effect of pulsed beam energies on the mean upper and bottom diameters of micropillars with a translation velocity of 1.0 m/s and a pulsed repetition rate of 600 kHz.

**Figure 6 materials-16-00244-f006:**
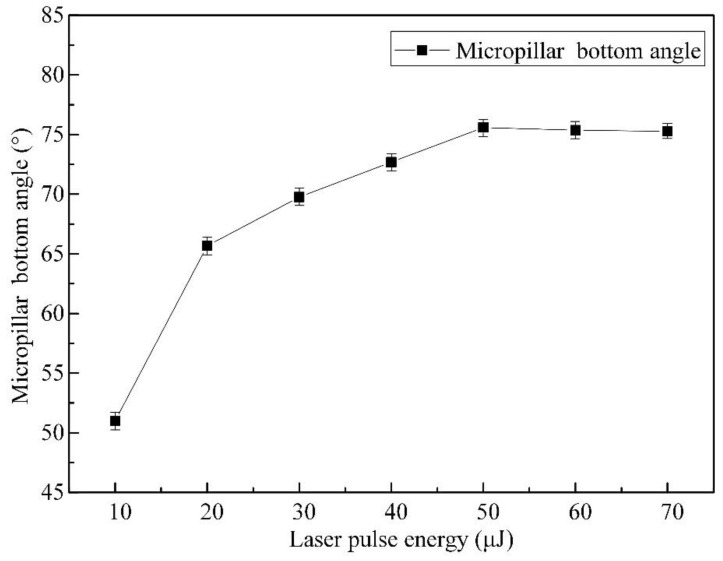
Effect of pulsed beam energies on the bottom angle of the micropillars with a translation velocity of 1.0 m/s and a pulsed repetition rate of 600 kHz.

**Figure 7 materials-16-00244-f007:**
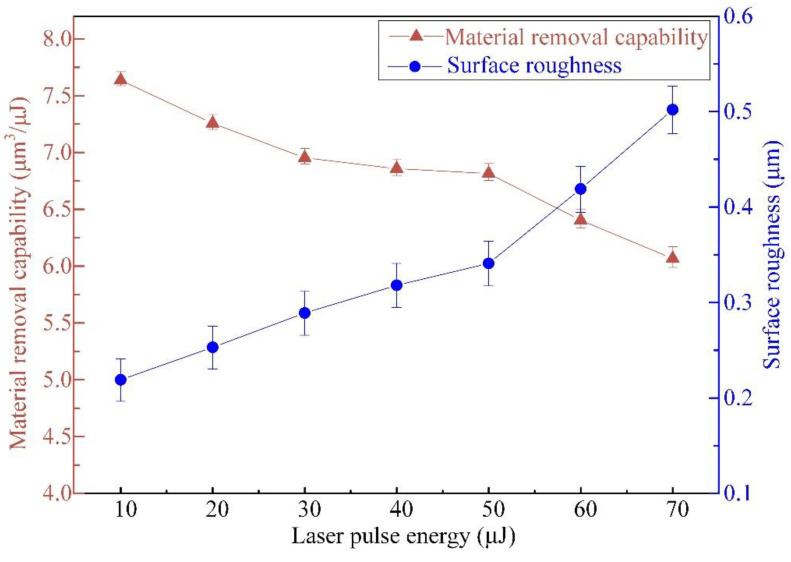
Effect of pulsed beam energies on the MRC and *R*a of micropillars at a translation velocity of 1.0 m/s and a pulsed repetition rate of 600 kHz.

**Figure 8 materials-16-00244-f008:**
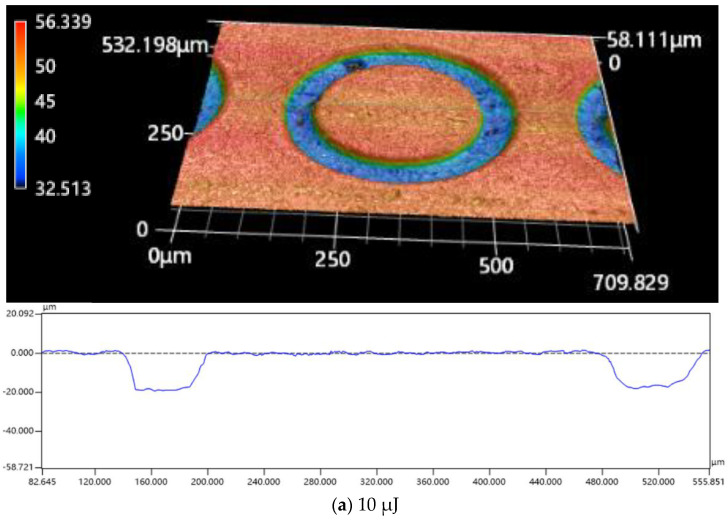
3 D microscope images and profiles of the micropillars obtained through various pulsed beam energies at a fixed translation velocity of 1.0 m/s and a pulsed repetition rate of 600 kHz: (**a**) 10 μJ; (**b**) 40 μJ.

**Figure 9 materials-16-00244-f009:**
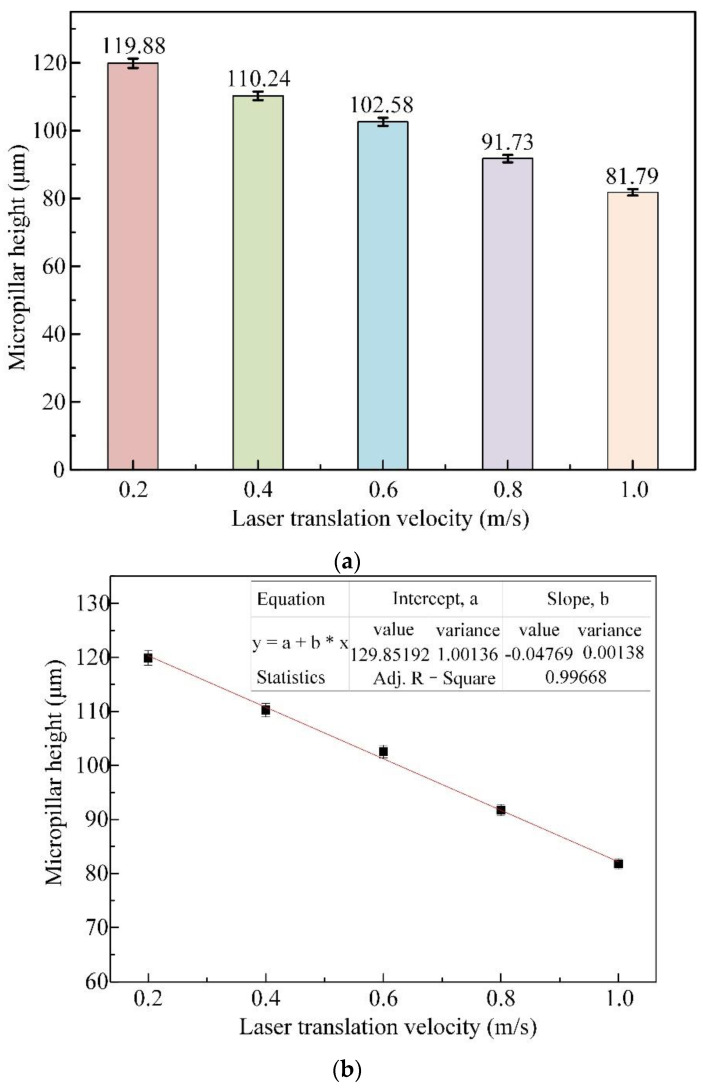
Influence of laser translation velocities on the mean height of micropillars, with a pulse energy of 50 μJ and a pulsed repetition rate of 600 kHz. (**a**) Measurements of micropillar height against laser translation velocity. (**b**) Linear fitting of micropillar height against laser translation velocity.

**Figure 10 materials-16-00244-f010:**
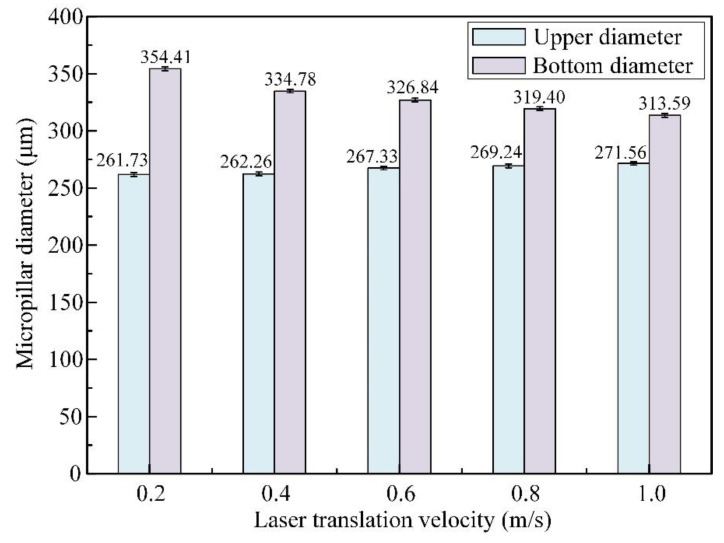
Influence of laser translation velocities on the average upper and bottom diameters of micropillars, with a pulse energy of 50 μJ and a pulsed repetition rate of 600 kHz.

**Figure 11 materials-16-00244-f011:**
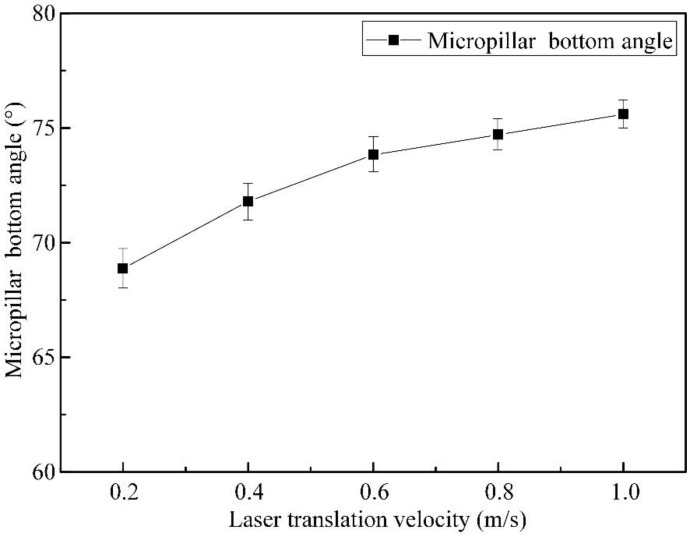
Influence of laser translation velocities on the bottom angle of micropillars, with a pulse energy of 50 μJ and a pulsed repetition rate of 600 kHz.

**Figure 12 materials-16-00244-f012:**
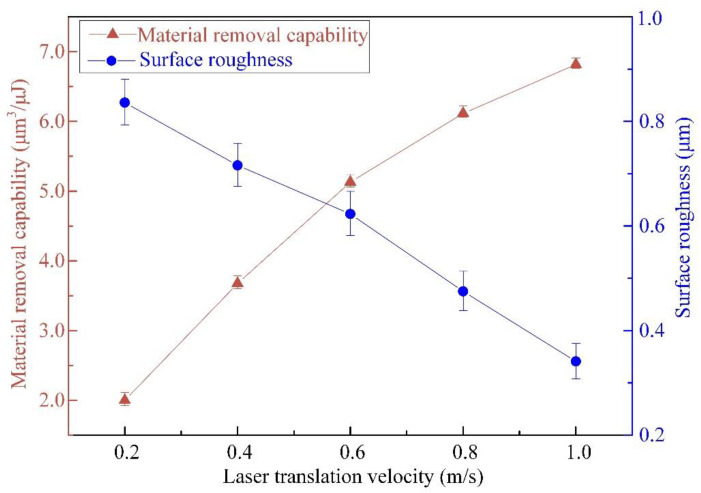
Effect of laser translation velocities on the material removal capability and surface roughness of micropillars, with a pulsed laser beam energy of 50 μJ and a pulsed repetition rate of 600 kHz.

**Figure 13 materials-16-00244-f013:**
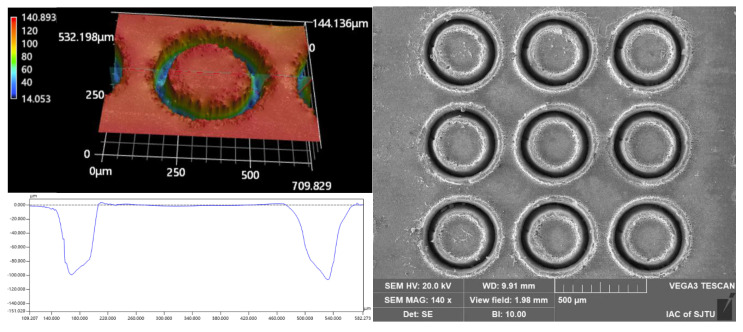
The cross-section profile and SEM image of the micropillars fabricated with a laser translation velocity of 0.2 m/s at a pulse energy of 50 μJ and a pulsed repetition rate of 600 kHz.

**Figure 14 materials-16-00244-f014:**
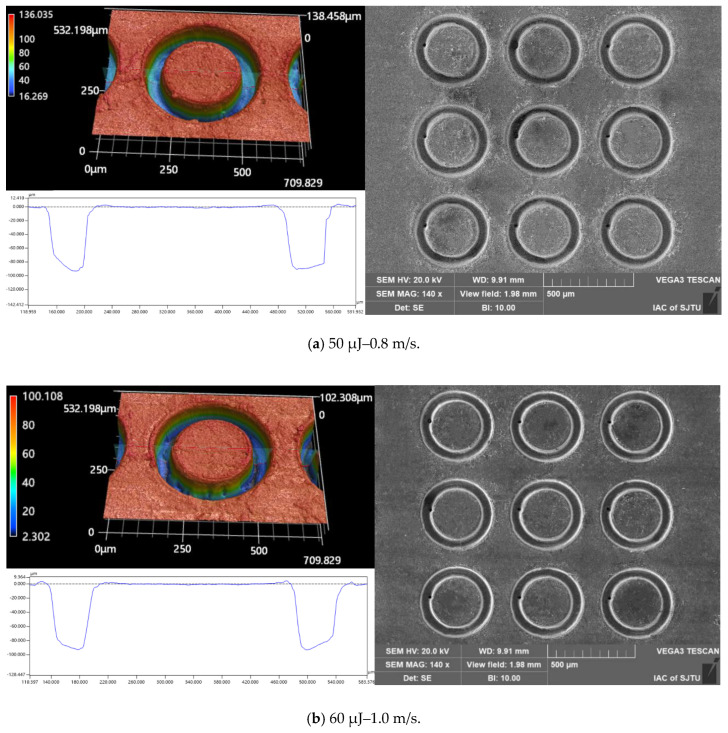
The cross-section profile and SEM image of the micropillars fabricated at the same height of 92 μm, with different laser machining parameters: (**a**) 50 μJ–0.8 m/s; (**b**) 60 μJ–1.0 m/s.

**Figure 15 materials-16-00244-f015:**
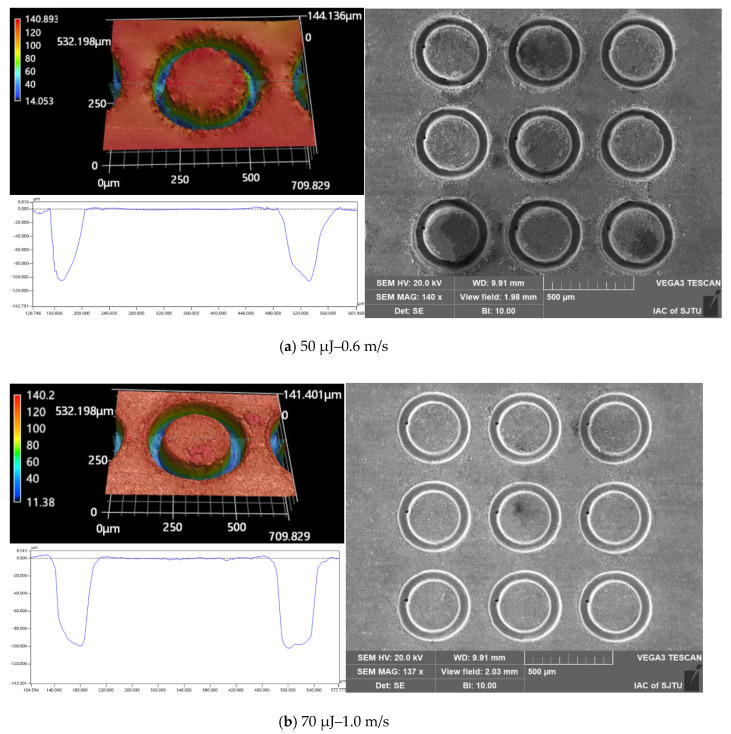
The cross-section profile and SEM image of the micropillars fabricated at the same height of 102 μm with different laser machining parameters: (**a**) 50 μJ–0.6 m/s; (**b**) 70 μJ–1.0 m/s.

**Table 1 materials-16-00244-t001:** The processing parameters with the femtosecond laser.

Parameter	Value
Wavelength	1028 nm
Pulse duration	218 fs
Pulse repetition rate	600 kHz
Laser spot radius	6 μm
Processing cycles	10
Scanning strategy	Spiral scanning
Translation velocity	0.2, 0.4, 0.6, 0.8, 1.0 m/s
Pulsed beam energy	10, 20, 30, 40, 50, 60, 70 μJ

**Table 2 materials-16-00244-t002:** Effect of different laser pulse energies and laser translation speeds on the micropillar bottom angle, MRC, and *R*a at the same micropillar height of 92 μm.

Laser Ablation Parameters	Micropillar Bottom Angle (°)	MRC (μm^3^/μJ)	*R*a (μm)
50 μJ–0.8 m/s	74.71	6.115	0.475
60 μJ–1.0 m/s	75.36	6.403	0.419

**Table 3 materials-16-00244-t003:** Effect of different laser pulse energies and laser translation speeds on the micropillar bottom angle, MRC, and *R*a at the same micropillar height of 102 μm.

	Micropillar Bottom Angle (°)	MRC (μm^3^/μJ)	*R*a (μm)
50 μJ–0.6 m/s	73.83	5.129	0.623
70 μJ–1.0 m/s	75.28	6.067	0.502

## Data Availability

The data that support the findings of this study are available from the corresponding author upon reasonable request.

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
