# Peer review of "Material Removal Capability and Profile Quality Assessment on Silicon Carbide Micropillar Fabrication with a Femtosecond Laser"

_materials, 2022, doi:10.3390/ma16010244_

Round 1
Reviewer 1 Report
The present work describes the process of micropillars on SiC using femtosecond 75 laser scanning in spiral mode. The impacts of pulsed beam energies and pulsed translation velocities on the micropillar profiles, dimensions, micropillar bottom angle, surface roughness, and material removal capability were examined.
The manuscript may be publishable in its current form.
However, Figures and Schemes are needs to improve.
The manuscript needs to be checked by a native English speaker.
There are a few grammatical errors are there.
In addition, a few recent reports are missing in the reference section.
Reviewer 2 Report
This article is quite well written but I fell that it is too much descriptive and that some efforts should be made to explain the observed phenomenon.
The introduction is well written and interesting but 22 references out of 24 if too much for me. Citing so many refences is useless and look like a way to impress, not to say overload with information, the reader but without really getting into the details.
Too much references in the intro and almost none in the other part.
For example some references about the experimental set up and the equation (1) would had been more useful than plenty of generic ones;
The second one is the lack of explanation of the observation made on some results.
Figure 6 is very soundful but I would like to have some explanation or at least some hypothesis made in order to understand why the saturation occurs. Same remarks for figure 12.
Reviewer 3 Report
Referee report on “Material removal capability and profile quality assessment on silicon carbide micropillars fabrication with femtosecond laser” by Xifang Zhang et al
Although this topic is of some interest, this manuscript in its present form cannot be recommended for publication and requires some improvement and clarification.
1. A clear disadvantage is the lack of understanding and specifics and what crystalline modification of silicon carbide is meant.
2. Furthermore, the introduction needs more general information about SiC and its important applications in optical devices, nanotechnology and nuclear material science. This is important to attract more reader interest and further incentive applications. For some of them, see, for example:
a) Huczko, A., Dąbrowska, A., et al . Silicon carbide nanowires: synthesis and cathodoluminescence. physica status solidi (b), 2009, 246(11‐12), 2806-2808.
b) Ning G., Zhang L., Zhong W., Wang S., Liu J., Zhang C. Damage and annealing behavior in neutron-irradiated SiC used as a post-irradiation temperature monitor
(2022) Nuclear Instruments and Methods in Physics Research, Section B: Beam Interactions with Materials and Atoms, 512 , pp. 91-95.
3. As mentioned above, it is well known that there are several crystalline modifications of SiC, and it is completely incomprehensible why this fact and the specification of a the crystal structure are not given and discussed in the text (lines 75 -78)?
4. There are several surface orientations of SiC. How do the discussed effects of the formation of the micropillars depend on this circumstance
5. Fig.4. This data looks strange, since such accuracy requires the indication of the measurement error and the corresponding description in the text.
6. Error bars and corresponding analysis are also needed for Fig. 5, 6, 8-12.
6. In the conclusions, it is necessary to clearly formulate what new data about the studied materials were obtained in this work?
In general, the manuscript is interesting and can be considered for publication after constructive reflection on the above comments.
Round 2
Reviewer 3 Report
The author have significantly improved their manuscript, which now can be recommended for publication.